# Waterborne Polyurethane Acrylates Preparation towards 3D Printing for Sewage Treatment

**DOI:** 10.3390/ma15093319

**Published:** 2022-05-05

**Authors:** Kunrong Li, Yan Li, Jiale Hu, Yuanye Zhang, Zhi Yang, Shuqiang Peng, Lixin Wu, Zixiang Weng

**Affiliations:** 1College of Chemistry, Fuzhou University, Fuzhou 350108, China; likunrong@fjirsm.ac.cn (K.L.); hujiale@fjirsm.ac.cn (J.H.); zhangyuanye@fjirsm.ac.cn (Y.Z.); 2Fujian Institute of Research on the Structure of Matter, Chinese Academy of Sciences, Fuzhou 350002, China; yangzhi@fjirsm.ac.cn (Z.Y.); pengshuqiang@fjirsm.ac.cn (S.P.); 3School of Ecological Environment and Urban Construction, Fujian University of Technology, Fuzhou 350118, China; yanli_amy@fjut.edu.cn

**Keywords:** waterborne polyurethane acrylate, 3D printing, sewage treatment, nitrification efficiency, direct ink writing

## Abstract

Conventional immobilized nitrifying bacteria technologies are limited to fixed beds with regular shapes such as spheres and cubes. To achieve a higher mass transfer capacity, a complex-structured cultivate bed with larger specific surface areas is usually expected. Direct ink writing (DIW) 3D printing technology is capable of preparing fixed beds where nitrifying bacteria are embedded in without geometry limitations. Nevertheless, conventional bacterial carrier materials for sewage treatment tend to easily collapse during printing procedures. Here, we developed a novel biocompatible waterborne polyurethane acrylate (WPUA) with favorable mechanical properties synthesized by introducing amino acids. End-capped by hydroxyethyl acrylate and mixed with sodium alginate (SA), a dual stimuli-responsive ink for DIW 3D printers was prepared. A robust and insoluble crosslinking network was formed by UV-curing and ion-exchange curing. This dual-cured network with a higher crosslinking density provides better recyclability and protection for cryogenic preservation. The corresponding results show that the nitrification efficiency for printed bioreactors reached 99.9% in 72 h, which is faster than unprinted samples and unmodified WPUA samples. This work provides an innovative immobilization method for 3D printing bacterial active structures and has high potential for future sewage treatment.

## 1. Introduction

The rapid development of the industrialization advancement of medicine and the chemical industry brings more discharged synthetic compounds into the water, especially the most common pollutant—ammonia nitrogen [1]. Ammonia nitrogen pollutants potentially threaten human beings’ health and fertility. By immobilizing bacterium in a fixed space, biological nitrification is deemed to be the most economical, green, and effective method to combat this [2,3]. However, continuous improvements are still required to develop fixation processes and carrier materials. It is significant to immobilize functionalized bacteria in nontoxic and robust carriers through mild and effective means to maintain the bacteria viability.

The conventional immobilized bacteria method represents fixing bacteria in limited regular shapes [4,5]. Nevertheless, it is hard to realize the importance of complex structures with a large specific area. Thus, the oxygen diffusion for bacteria reproduction is impeded. Three-dimensional printing technologies can process complex three-dimensional structures without geometry limitations [6,7]. Although the fused deposition modeling (FDM) [8,9] and stereolithography (SLA) [10,11] methods are being used successfully in a number of applications, they are not suitable for printing bioactive inks because their printing process is not friendly to bacteria [12,13]. Recently, the emergence of direct ink writing (DIW) 3D printing technology has provided an alternative way to prepare complex-structured fixed beds for bacteria cultivation and presents the most mild processing condition by minimizing the collateral damage to bacteria [14,15,16]. The advances of the DIW 3D printing technique make itself capable of being implemented in biological-related fields, including biosensing, tissue regeneration, environmental sensing, drug discovery, and clinical application [17,18,19,20,21,22]. The DIW 3D printing technique employs a pressure-driven extrusion method to prepare parts instead of melting extrusion, which is more friendly to entrapped bacteria. Yet, these unique application scenarios place higher requirements on corresponding material development, including better biocompatibility and a proper rheology property—a shear thinning behavior for the employed material, specifically [23].

Nevertheless, conventional carrier materials used in sewage treatment such as agar, gelatin and polyvinyl alcohol (PVA) are poorly printable for DIW. Benefiting from its exclusive toughness and low toxicity, waterborne polyurethane (WPU) has been widely employed in 3D printing [24]. WPUs exhibit various mechanical properties by tailoring polymer chain structures, including chain extenders, polyols, and diisocyanate. This characteristic property enables WPUs to be widely implemented in extrusion-based 3D printing [25,26]. Most of the reported WPUs used in DIW were always reinforced by other materials to enhance the printability [27]. As for bacterial immobilization materials, water insolubility, biocompatibility, mechanical stability and non-biodegradability are needed for sewage treatment application [28,29,30]. Based on this, WPUs must be modified to obtain excellent performance. It has been reported that the bacteria were safe under the exposure of UV light with a wavelength of 365 nm [31]. The combination of UV and ion-exchange curing delivers a more bio-friendly fabrication process, facilitating bacteria viability. Sodium alginate (SA) is a natural macromolecule that can undergo ion exchange with Ca^2+^. It can be used as a potential ion-exchange curing agent, which also crosslinks with WPUs to further improve the mechanical properties, rheology, and biocompatibility of the hybrid system.

This work aims to prepare an immobilized material with high tensile strength and strain, outstanding stability in practical usage, excellent biocompatibility to nitrifying bacteria and proper shear thinning behavior for DIW 3D printing. The ink preparation method is illustrated in Figure 1. First, a WPUA modified by amino acid (L−Lysine) was synthesized. Here, the introduction of L−Lysine enhanced the mechanical property of WPUA and provided a milder bio-environment for bacterial cultivation [32,33]. Subsequently, to improve the printability of the material, SA was used to increase the zero-shear viscosity of the ink due to the formation of hydrogen bonds [34,35,36]. Before being adopted in a DIW 3D printer, the rheology properties were first studied to evaluate the suitability for 3D printing. From the results of the rheology study, proper extrusion pressure, diameter of the nozzle and the layer thickness were adjusted to print as-prepared WPUA to a bioreactor with a complex structure. During the printing process, a water insolubility initial crosslinking network was formed via UV curing. Subsequently, a final robust crosslinking network was formed via ion-exchange curing. Additionally, the reusability of the printed bioreactors was tested using cycling experiments. At last, to determine the protective effect of the material against nitrifying bacteria at low temperatures, we also simulated refrigerated storage experiments.

## 2. Materials and Methods

### 2.1. Materials

In this work, a polyols mixture comprising poly-caprolactone glycol (PCL) (Mn = 2000 g mol^−1^) and polyethylene glycol (PEG) (Mn = 2000 g mol^−1^) was employed. Isophorone diisocyanate (IPDI), 2,2-bis(hydroxymethyl) propionic acid (DMPA), ethylenediamine (EDA) and L−Lysine (L−Lys) were employed. Tin(II) 2-ethylhexanoate (Sn(Oct)_2_) as a catalyst, 2-hydroxyethyl acrylate (HEA) as a blocking agent, triethylamine (TEA) as a neutralizer and methyl ethyl ketone (MEK) as a solvent were used. In the WPUA system, sodium alginate (SA) with a molecular weight of 32,000–250,000 was added for the formation of an interpenetrating polymer network (IPN), and calcium dichloride (CaCl_2_) was used as a cross-linking agent. The abovementioned reactants were all purchased from Aladdin Industrial Co., Ltd., Shanghai, China. In this work, water-soluble photoinitiator 2,2′-azobis [2-methyl-N-(2-hydroxyethyl) propionamide] (VA−086) was purchased from Heowns, Tianjin, China. The heterotrophic nitrifying bacteria were cultivated by the Institute of Urban Environment, Chinese Academy of Sciences.

### 2.2. Preparation of WPUA Emulsion and Ink Matrix

The synthetic route of WPUAs is shown in Figure 2. Here, a mixture of PCL and PEG (weight ratio = 19:1) was introduced in a three-neck, 250 mL bottle and mechanically stirred at 75 °C in a nitrogen atmosphere. Afterward, IPDI mixed with 0.05 wt.% of Sn(Oct)_2_ was added dropwise and kept reacting at 75 °C. The extent of the reaction was monitored with Fourier transform infrared spectroscopy. Subsequently, after increasing the temperature to 90 °C, DMPA and MEK were added to decrease the viscosity of the system from 350.37 Pa s to 109.23 Pa s. To terminate the chain condensation, HEA was added dropwise to graft UV-curable acrylate functional groups at 75 °C. This capping reaction was monitored using FTIR. To neutralize the remaining carboxyl functional groups introduced by DMPA, TEA was added at 40 °C and stirred for another 30 min.

The two chain extenders (EDA and L−Lys) were dissolved in deionized water and added to the oligomer system dropwise with vigorous stirring (1200 rpm) for 40 min. Then, a solid content of 30% WPUA emulsion was obtained. WPUAs synthesized with EDA and L−Lys as chain extenders were named WPUA−E and WPUA−L, respectively. The stoichiometric ratios of reactants are listed in Appendix A. At last, photoinitiator VA−086 (2 wt.% to solid content) was added and mixed uniformly. The remaining MEK and TEA were removed with a rotary evaporator.

A 10 wt.% of SA solution was first prepared to obtain DIW ink. Then, different mass ratios (1/10, 2/10, 3/10 and 4/10) of as-prepared SA solution and WPUA emulsions were prepared with a planetary centrifugal mixer for 30 min at a rotation speed of 1500 rpm.

Films were prepared as follows: various emulsions were poured into a Teflon-made mold. All samples were dried at room temperature for 2 days. The mold was then moved to a UV-curing oven (Dongguan, China) (5 W, 385 nm) for curing. The distance between the light source and the sample was 5 cm.

### 2.3. Cultivation of Heterotrophic Nitrifying Bacteria

The heterotrophic nitrifying bacteria were isolated from a sewage treatment plant. The corresponding formulation of artificial sewage is shown in the Appendix A. Subsequently, individual colonies were transferred to a synthetic sewage culture medium. The heterotrophic nitrifying bacterial culture medium was transferred every 48 h.

### 2.4. DIW 3D Bioprinting

Cultivated heterotrophic nitrifying bacteria were employed for further bio-ink preparation. In total, 100 g of cultivated bacteria medium was first centrifuged at 8000 rpm for 5 min. Bio-ink was obtained by mixing the bacterial concentrate with a 50 g ink matrix. A homemade DIW 3D printer (Dongguan, China) (Appendix A) attached with a UV curing spotlight accessory was created for further scaffold preparation. The movement speed of the syringe was 6 mm/s, and the platform was heated to 30 °C. The air pressure for pumping out the bio-ink was set to 60–90 kPa. The diameter of the syringe needle was 0.61 mm. According to our previous experiments, the layer height was expected to be smaller than the diameter of the syringe needle to achieve a high printing accuracy. In this work, we set the layer thickness as 0.58 mm. The power of the UV light was 5 W at a wavelength of 385 nm, and the distance between the light source and the pin head was around 5 cm. In this work, we drew models with CAD software and imported them into the DIW 3D printer. By adjusting the filling rate (20%, 30% and 40%, respectively, in this work) and infill patterns, scaffolds could be prepared (Figure 3) [37,38,39,40,41,42,43].

Subsequently, samples were transferred to a CaCl_2_ solution at a concentration of 4 wt.% at room temperature for 1 h for ion exchange. Heterotrophic nitrifying bacteria in the robust samples were further cultivated in artificial sewage with a shaker. The bacteria were activated for 12 h at 30 °C with oscillation.

After the activation, all samples were transferred to new artificial sewage in a weight ratio of sample: sewage = 5:100. The ammonium ion concentration of the supernatant was measured every 12 h. The bacterial viability of scaffolds was evaluated every 24 h.

### 2.5. Cyclic Processes and Simulation of Refrigerated Storage Experiment

To test the stability of materials in removing ammonia nitrogen from wastewater, another five consecutive cycles experiments were further carried out. After the cycle experiments, the last cycle samples were sealed in centrifuge tubes without sewage at 4 °C for 5 d, 7 d and 14 d, respectively. Samples were transferred into artificial sewage again and, ammonium ion concentration was analyzed every 12 h.

### 2.6. Characterization

#### 2.6.1. Physical–Chemical Characterization

Attenuated total reflectance Fourier transform infrared (ATR-FTIR) spectra of the dried WPUA film were obtained using a Nicolet 6700 with an ATR accessory (Thermo Fisher, Waltham, MA, USA). The wavelength ranged from 4000 to 400 cm^−1^. The spectra of the samples were obtained after 16 scanning times. The ^1^H NMR spectra were obtained on a Bruker-BioSpin Avance III 400 MHz spectrometer (Bruker BioSpin, Switzerland) using tetramethylsilane (TMS) as an internal standard and deuterated chloroform-d (CDCl_3_) as a solvent.

A rheometer (DHR-2, TA Instruments, New Castle, DE, USA) was used to study the rheology behavior. For viscosity study, steel-made parallel geometries (*ϕ* = 40 mm) were employed. Static measurement was carried out, and the shear rate ranged from 0.01 to 100 s^−1^. The storage shear modulus (*G*′) and loss shear modulus (*G*″) were studied through the oscillatory amplitude sweeping method, and the strain started from 0.01 to 1000% at an angular frequency of 1 rad/s. The hysteresis behavior of the ink was characterized with flow experiments setting up to two stages with shear rates from 0.5 to 100 s^−1^ and then from 100 to 0.5 s^−1^. To investigate the rheological properties of the ink during DIW printing, a simulation experiment including a high shear rate (80 s^−1^) and a low shear rate (0.01 s^−1^) were carried out to simulate the shear forces during the printing process, with a setting time of 60 s for each procedure.

The water absorption rate of UV-cured films in distilled water was estimated using the swelling method. The films were cut into sheet film at 10 mm × 10 mm × 0.5 mm and weighted (*w*_1_). The samples were put into deionized water under 30 °C and taken out after 12 h, 24 h, 48 h and 72 h, respectively. The films were dried using filter papers to remove surface water and weighed (*w*_2_). The water absorption was calculated using Equation (1).
(1)$$\mathrm{Water}\;\mathrm{absorption}\;\left(\%\right)=\frac{w_{2}-w_{1}}{w_{1}}\times 100$$

A thermal gravimetric analyzer (TGA, STA449C, Netzsch, Selb, Germany) was used to evaluate the thermal properties of films. The temperature ranged from 30 to 800 °C at a heating rate of 10 °C min^−1^ under a nitrogen atmosphere. Differential scanning calorimetry (DSC 25, TA Instruments, New Castle, DE, USA) was performed to calculate the glass transition temperature (*T*_g_) of the film. Each sample (about 10 mg) was put into an aluminum pan and heated from −80 to 20 °C with a heating rate of 10 °C min^−1^ under nitrogen purging.

The mechanical properties of samples, including the tensile property of films and the compression performance of printed scaffolds, were all tested with a universal material test machine (AGX-100 plus, Shimadzu, Kyoto, Japan). Films were cut into a sheet shape with a size of 25 mm (length) × 5 mm (width) × 0.5 mm (thickness). The tensile property of the films was examined with a stretching speed of 10 mm/min. A compression test for scaffolds was run at a compression speed of 5 mm/min.

A scanning electron microscope (SEM, Hitachi, SU8010, Tokyo, Japan) was used to observe the microstructure at the cross-fracture section and the surface of samples before and after DIW 3D printing.

#### 2.6.2. The Biological Study of Nitrifying Bacteria Growth in the Scaffold and Corresponding Nitrification Effect

Before counting, samples were first crushed in a sterile environment and diluted with PBS solution. The number of colonies per gram of the sample was calculated using the plate counting method. The initial number of bacterial colonies per gram of sample was first calculated via direct plate counting of the freshly mixed bio-ink (*n*_0_). After printing, the samples at a particular time—0 h, 24 h and 48 h, respectively, were taken out, and the number of bacterial colonies was counted as *n*_1_. Finally, we evaluated the viability of heterotrophic nitrifying bacteria with Equation (2). The average number from three samples was used to calculate the results of each experiment.
(2)$$\mathrm{Bacterial}\;\mathrm{Viability}\;\left(\%\right)=\frac{n_{1}}{n_{0}}\times 100$$

In addition, for a better intuitive observation of the microbial survival, a Zeiss LSM710 CLSM confocal laser scanning microscope (Zeiss, Oberkochen, Germany) (excitation: 488 nm; emission: 543 nm) was used. Before the observation, the bacteria were first stained with LIVE/DEAD^®^ BacLightTM Bacterial Viability kits (Invitrogen, Carlsbad, CA, USA).

The ammonia nitrogen measurement method of sewage used referred to the previous reports with an improvement by reducing the usage of samples [44,45]. The data were collected with an ultraviolet spectrophotometer (PerkinElmer, Lambda 950, Waltham, MA, USA) and determined by the average number from three samples. The method details are provided in Appendix A. The final nitrification efficiency of samples was calculated with Equation (3).
(3)$$\mathrm{Nitrification}\;\mathrm{Efficiency}\;\left(\%\right)=\frac{c_{0}-c_{1}}{c_{0}}\times 100$$
where *c*_0_ and *c*_1_ represent the initial ammonia nitrogen concentration and the concentration after specific time samples, respectively.

## 3. Results and Discussion

### 3.1. Characterization of WPUAs

The physicochemical properties of as-synthesized WPUAs were studied. First, FTIR was used to verify the chemical structure of synthesized WPUA. The results are shown in Appendix A. The a, b and c steps in Appendix A stand for a, b and c steps in Figure 2, respectively. It can be seen that the intensity of the -NCO peak that appeared in the range of 2280–2240 cm^−1^ decreases continuously as the reaction progress, and the corresponding peak area of the −NCO group in Appendix A decreases gradually. The results for FTIR spectra confirmed the optimal reaction time for each step, which, corresponding to a, b, and c steps, were 3 h, 3 h and 4.5 h, respectively. The final chemical structures of these two types of WPUAs were confirmed with the FTIR spectrum and ^1^H NMR spectrum. Appendix A represents the FTIR spectra of two samples. In the spectra, the peak observed in 3370 cm^−1^ is related to the stretching vibration of the N-H bond in secondary amine, and peaks at 1726 cm^−1^ belong to the C=O functional group in esters and carboxyl. The methylene peaks at 2947 cm^−1^ and 2869 cm^−1^ correspond to asymmetric stretching vibration peaks and symmetric stretching vibration peaks. The peaks showed in 1654 cm^−1^, 1418 cm^−1^ and 810 cm^−1^ represent the UV-curable functional groups (C=C, =CH_2_ and =CH, respectively), indicating that the UV-curable groups were successfully incorporated into the WPU polymer chain. Meanwhile, no characteristic peaks belonging to −NCO between 2280 cm^−1^ and 2240 cm^−1^ were observed in the FTIR spectrum, indicating a complete reaction for -NCO in the WPUA system. Additionally, the structures of WPUA were confirmed with the results from ^1^H NMR (Appendix A). Three peaks at 6.4 ppm (a), 6.1 ppm (b) and 5.8 ppm (c) were attributed to the −CH=CH_2_ from HEA units. The peaks at 4.0 ppm (e), 1.6 ppm (i), 2.9 ppm (g) and 2.3 ppm (h) were assigned to the CH_2_ from PCL units. The 3.6 ppm (f) peak was attributed to the CH_2_ group in PEG and IPDI units [46].

### 3.2. Rheology and Water Swelling Properties

For DIW 3D printing, utilized bio-ink is expected to exhibit a shear-thinning behavior for better free-standing upon extrusion. The viscosity versus shear rate variation curves for WPUA−E and WPUA−L samples with SA are shown in Figure 4A,B, respectively. From Figure 3, it can be found that the introduction of the SA solution leads to an increase in the viscosity of the system. Meanwhile, a more obvious shear thinning behavior was observed. It can be seen from the figure that when the mass ratio of 10 wt.% SA solution and WPUA−E reached 1/10, and the viscosity increased from 595.35 Pa s to 4778.75 Pa s (Figure 4A). The steeper curves observed in Figure 4A,B proved that the introduction of SA solution exacerbated the shear-thinning behavior. This increase was also attributed to the formation of hydrogen bonds between hydroxyl in SA and carbonyl in WPUA and the hydrogens bonds between carbonyl in SA and secondary amine in WPUA [47]. From Figure 4A, it is worth noting that when the mass ratio of 10 wt.% SA reached 2/10, the highest zero-shearing viscosity was obtained (around 8650.48 Pa s) due to the formation of the solution and WPUA−E hydrogen bonds. Nevertheless, a further increase in SA solution content resulted in a decrease in viscosity due to the reduction in hydrogen density in the system. At a large shear rate range (100 s^−1^), the viscosity of all samples tended to be the same (around 10 Pa s). This is due to the breakage of the formed hydrogen bonds by SA solution, which leads to a severe viscosity decrease. This indicates that the introduction of the SA solution is conducive for DIW 3D printing. Because both figures showed that the 2/10 sample had the most significant zero-shear viscosity (8650.48 Pa s and 15,003.80 Pa s), we chose the 2/10 samples in Figure 4A,B as the subsequent DIW printing inks and named them WES and WLS, respectively.

Oscillation amplitude sweeps show that the four samples, WPUA−E, WPUA−L, WES and WLS, exhibited mainly elastic behavior at low strains and apparent viscous behavior at high oscillation strains (Figure 5A). It should be noted that the moduli for WES and WLS samples were more sensitive to the variation of the strain. After the strain increased to 10%, the storage modulus decreased dramatically with the stress, exhibiting significant viscous behavior. In comparison, the gel–sol conversion points of WPUA−E and WPUA−L samples were around 73% strain.

Additionally, we evaluated the hysteresis behavior–a critical factor for controlling the print quality of the inks by flow experiments with two stages. Commonly, the smaller the area of the closed loop, the lower the hysteresis. In Figure 5B, the small closed-loop area shows that as-prepared samples are capable of DIW 3D printing with high accuracy.

Meantime, curves representing shear rate versus viscosity shown in Figure 6 demonstrate how as-prepared inks behave under the shear rate change. From Figure 6, one can notice that as-prepared WES and WLS responded quickly to the variation of the shear rate, and no time latency can be observed. This fast response character avoids inks leaking during the movement of the syringe and guarantees precision printing. It can be concluded that the introduction of SA makes WPUA more suitable for DIW printing. In addition, L−Lys modified WPUA−L had higher zero-shear viscosity after mixing with SA, because L−Lys brought more hydrogen bonding sites to WPUA. When SA was added, the ink formed more hydrogen bonds. By simulating the DIW printing process, WLS exhibits better printing performance.

As a carrier material for biological nitrification, it is essential to evaluate the swelling performance of the bacterial carrier [48,49,50]. Figure 7 shows the comparison curves of the swelling performance between WPUA−E, WPUA−L, WES and WLS. It can be seen that water absorption decreases in the order of WPUA−L > WPUA−E > WLS > WES. A slightly greater water absorption of WPUA−L film than WPUA-E film was observed due to the fact that the amino acid caused more strong hydrophilic groups, carboxyl groups. Due to the introduction of SA, a proper cross-linking system was formed, and the water absorption of WES and WLS was improved. The water absorption rates in this work were acceptable for immobilized nitrifying bacteria technology.

### 3.3. Mechanical Properties

In this work, as-synthesized emulsions were cast as films to evaluate the tensile strength with a universal test machine. All curves are compiled in Figure 8A. Corresponding results showed that all as-prepared samples exhibit typical elastomer tensile behavior with the emergence of a yield point. Specifically, WPUA-L exhibits the most outstanding mechanical properties among all samples, with a tensile strength of 7.1 MPa and an elongation at a break of 350%. This is because semi-crystallinity phase separation occurred in the WPUA system [51]. Such outstanding mechanical properties promise the durability of the bacteria carrier, providing a longer service life. Additionally, corresponding compression tests were performed, and data are compiled in Figure 8B. Compared with traditional nitrifying bacteria immobilization materials such as PVA, PVA/SA, and pure SA, in this work, WPUA-L has better mechanical properties, indicating a higher stability of WPUA-L for sewage treatment [52]. The compression tests show that the as-prepared bacterial carrier with a lattice structure exhibits outstanding stress tolerance. During the whole compress procedure, no cracks were observed from the smooth surface of the curve. Additionally, the relationship between the infill ratio and the stability was investigated. Thanks to the outstanding mechanical properties of cured WPUA, no cracks were observed no matter what the in-fill ratio was. The as-printed samples at the fill rate of 20%, 30% and 40% are shown in Figure 8A. According to the above research, we chose the fill rate of 30% samples as the scaffolds for testing biomedical properties. Finally, we changed the infill pattern and printed three precise structures, as shown in Figure 9B. It can be seen that the bioreactors have good mechanical properties even in different fill patterns.

### 3.4. Biomedical Properties Evaluation

The nitrification efficiency of different materials is shown in Figure 10A. Initially, both samples were sterilized first to eliminate the errors caused by the materials. The sterilized samples proved that the nitrifying effects were caused by bacteria instead of materials. Following this, both lattice samples carrying nitrifying bacteria prepared with WES and WLS, respectively, were soaked in the artificial sewage to evaluate the nitrification effects. The results showed that both bacteria-loaded, 3D-printed samples based on WES and WLS achieved around 99.9% ammonia nitrogen removal within 96 h, indicating that WES and WLS are promising candidates as bacteria-carrying materials. Significantly, the printed, WLS-loaded bacteria group displayed the most outstanding nitrification performance, proving that the WLS material is more friendly for nitrifying bacteria. For a better comparison, the nitrifying effect caused by unprinted WES samples carrying bacteria was studied (green line in Figure 10A). The results showed that as-printed samples exhibit better nitrification efficiency. This can be attributed to the higher specific surface area of printed samples to ensure the flow of oxygen and nutrients required for the survival of nitrifying bacteria.

In addition, the survival rate of nitrifying bacteria was quantitatively evaluated using the plate count method (Appendix A), and the results are shown in Figure 10B. The trends of growth of bacteria and nitrification efficiency coincided. Although, the printed sample groups (WES and WLS after printing) lost 28.53% and 30.55% of bacteria, respectively. While unprinted WES lost only 9.5% of bacteria due to UV curing, the number of colonies per unit weight of the printed samples after 24 h was much higher than that of the unprinted WES group. Specifically, the number of bacteria in the WLS-printed group increased up to 2564.17% at 72 h, followed by the WES printed group at 1748.18%, and the WES unprinted group had the smallest increase of 598.01%.

To evaluate bacteria viability more intuitively, the WLS with printing samples were further observed with a confocal laser scanning microscope; the results are shown in Figure 11. It can be seen that the red fluorescence representing inactive bacteria is inconspicuous at 0 h, 24 h, 48 h and 72 h. In comparison, the green fluorescence representing active bacteria is obvious at 24 h, 48 h and 72 h. This indicates that the nitrifying bacteria maintain their microbial activity during the printing process owing to the material providing a favorable medium for the growth and reproduction of nitrifying bacteria.

To evaluate the recyclability of samples, we selected the printed WLS with the fastest nitrification efficiency and conducted another five cycles of experiments. Figure 12A represents the nitrifying effects comparison at different cycles. It can be found that as the number of cycles increases, the time taken to reach 99.9% nitrification efficiency decreases, which is due to the fact that nitrifying bacterial reproduction in the WLS material and the number of bacteria was increased, improving the nitrogen removal efficiency. It should be noted that a better nitrification performance than free bacteria was observed at the sixth cycle, reaching 99.9% of ammonia nitrogen removal efficiency in only 36 h. The above results show that the prepared WLS carriers have a good biocompatible performance along with a good water resistance and mechanical properties, ensuring repeated employment for multiple cycles of experiments.

We also conducted experiments simulating refrigerated transport, and the results are shown in Figure 12B. It can be seen that the nitrification curves of refrigeration for 5 d, 7 d and 14 d almost overlapped. All samples reach a high removal rate at about 36 h, indicating that long-term refrigeration storage has little effect on the activity of nitrifying bacteria, proving that the WLS material has an excellent protective effect on nitrifying bacteria at low temperatures and can be transported while refrigerated without cultivating medium.

To study the mechanism of enhancement by 3D printing, SEM images were captured to observe the microscopic morphology of the samples (Figure 13). The 3D-printed sample has a denser microporous structure inside. Heterotrophic nitrifying bacteria are firmly fixed in the material, and it can be seen that the material can effectively prevent the loss of bacteria. With comprehensive analysis, better nitrifying effects caused by DIW 3D printing can be explained by the fact that the specific surface area was much smaller compared to the printed samples, which restricted the circulation of artificial wastewater, and nitrifying bacteria were not supplied with sufficient ammonia nitrogen and organic carbon sources, resulting in lower nitrification efficiency.

## 4. Conclusions

In this work, a dual stimuli-responsive, as well as light-curing and ion-exchange-curing, DIW 3D printing ink (WLS) composed of L−Lys-modified WPUA and SA was prepared. This ink exhibits excellent biocompatibility, providing a bio-friendly medium for the growth and reproduction of nitrifying bacteria. Additionally, as-prepared WLS has excellent recyclability and delivers outstanding protection for bacteria at low temperatures. In addition, we found that the samples printed via DIW exhibit better nitrification efficiency because of the existence of a larger specific surface area. All of the above experiment results show that as-prepared WPUA modified by L−Lys and SA composite is a promising candidate material for DIW 3D printing for the application of biological nitrification and sewage treatment.

## Figures and Tables

**Figure 1 materials-15-03319-f001:**
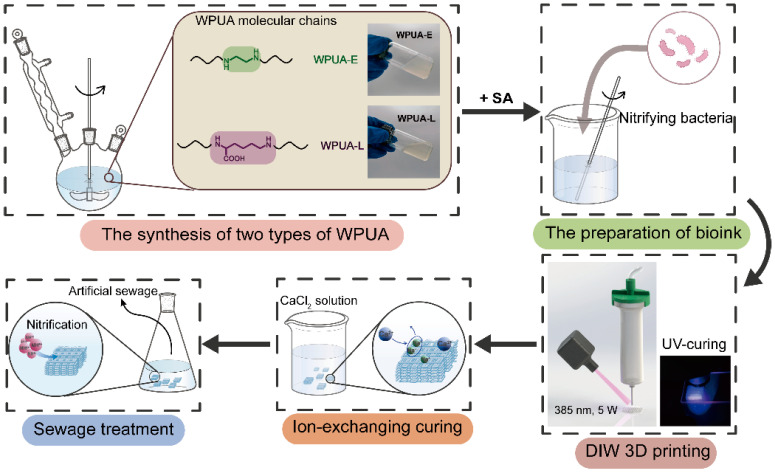
The synthesis, bioprinting and application of WPUA in sewage treatment.

**Figure 2 materials-15-03319-f002:**
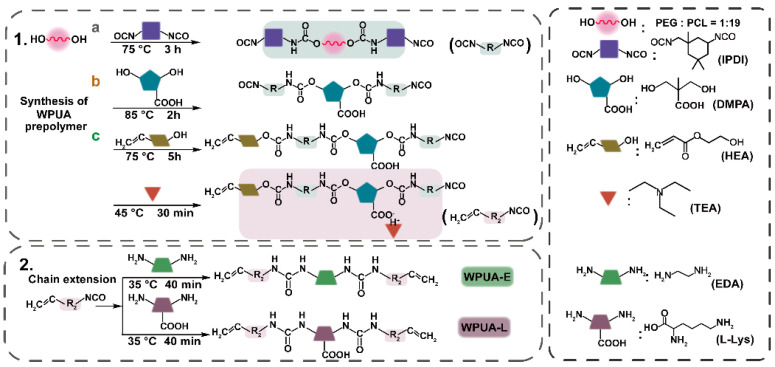
Synthetic route of two WPUAs with EDA and L−Lys as chain extenders.

**Figure 3 materials-15-03319-f003:**
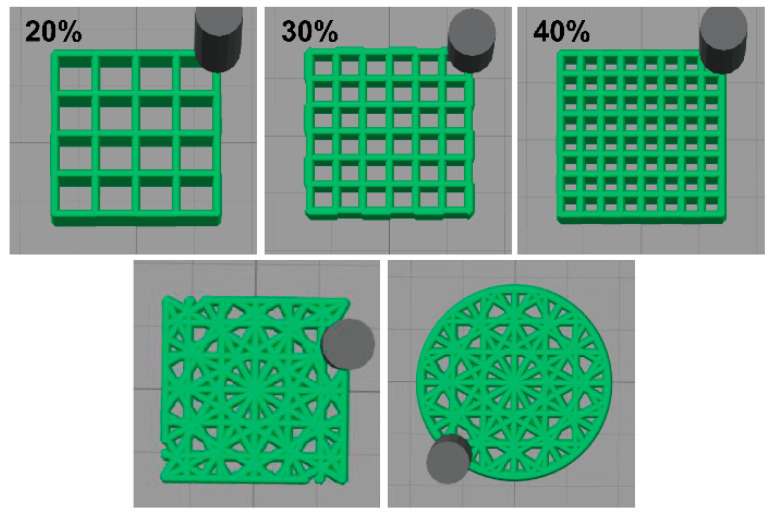
A series of 3D scaffolding slice models with different infill rate and infill patterns were prepared with Simplify3D software.

**Figure 4 materials-15-03319-f004:**
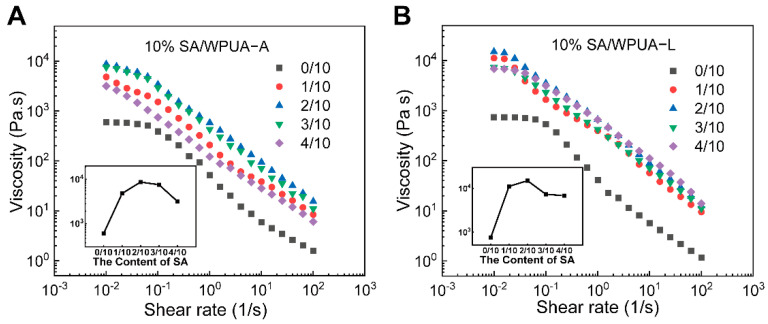
The steady-state flow behavior of (**A**) WPUA−E and (**B**) WPUA−L composite inks were measured with viscosity curves.

**Figure 5 materials-15-03319-f005:**
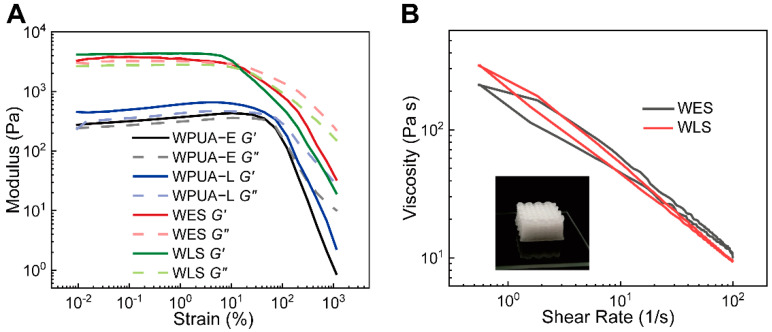
(**A**) Oscillatory amplitude sweeps (strain, 0.01 to 1000%; angular frequency, 1 rad/s) of emulsions. (**B**) Hysteresis curves of WES and WLS at shear rate 0.5–100 s^−1^.

**Figure 6 materials-15-03319-f006:**
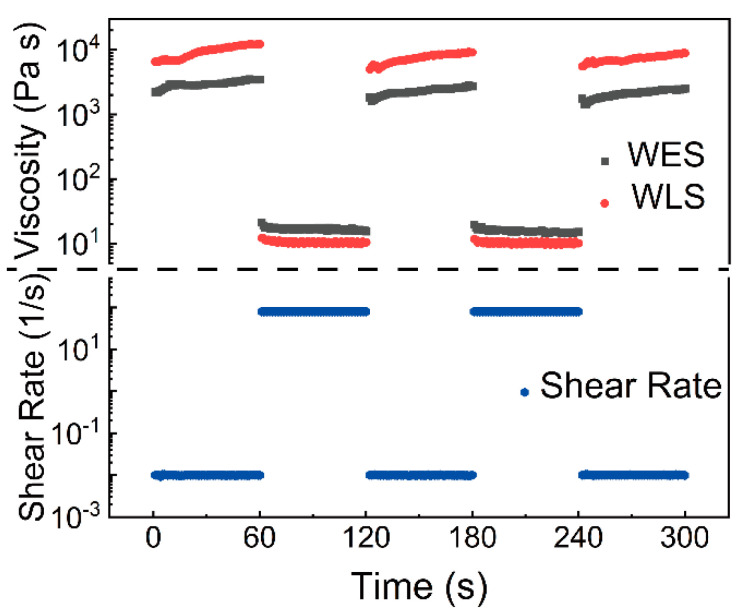
Viscosity variation of WES and WLS in DIW 3D printing condition simulation.

**Figure 7 materials-15-03319-f007:**
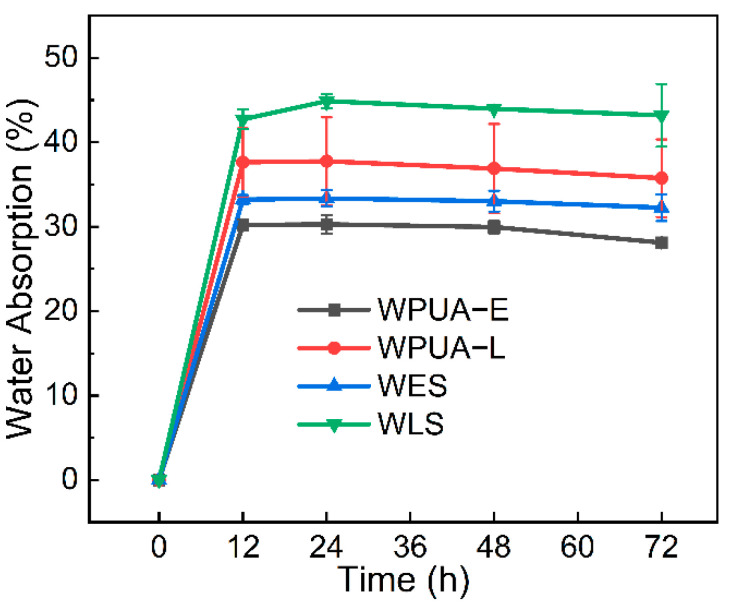
The water absorption of WPUA films.

**Figure 8 materials-15-03319-f008:**
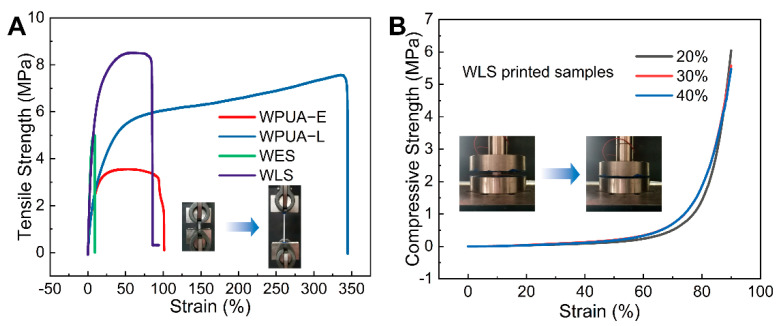
(**A**) Tensile properties of WPUA−A, WPUA−L, WES and WLS films. (**B**) Compression properties of WLS samples with 20%, 30% and 40% filling ratio, respectively.

**Figure 9 materials-15-03319-f009:**
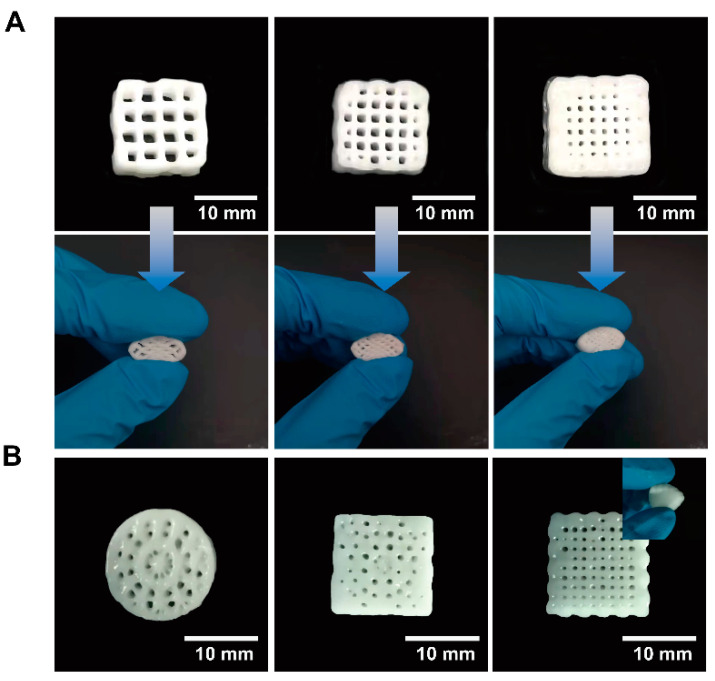
(**A**) Photos of compression test samples. (**B**) The printing bioreactors with other infill patterns.

**Figure 10 materials-15-03319-f010:**
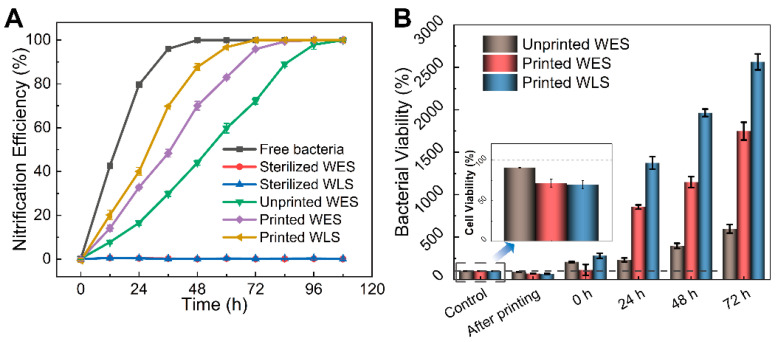
(**A**) Nitrogen removal effects of different materials. (**B**) Bacterial viability of printed WES, unprinted WES and printed WLS sample groups.

**Figure 11 materials-15-03319-f011:**
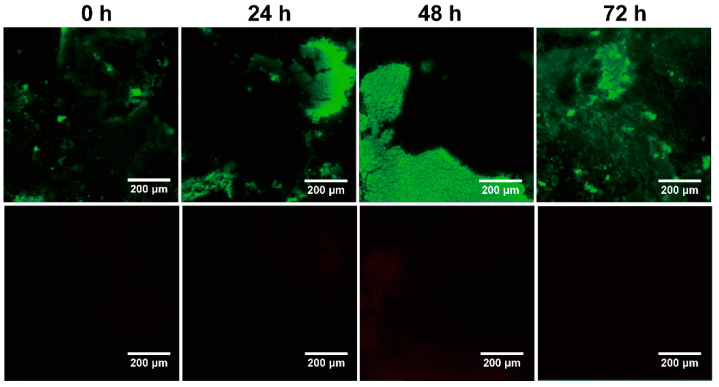
Fluorescence images of WLS samples were observed at 0 h, 24 h, 48 h and 72 h, respectively.

**Figure 12 materials-15-03319-f012:**
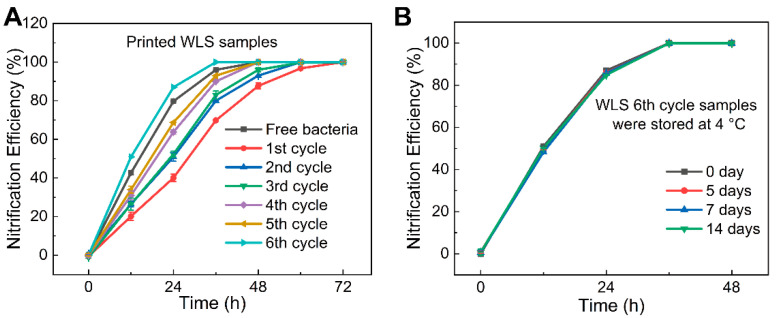
(**A**) The nitrogen removal cycle experiment of WLS samples. (**B**) The simulation of refrigerated storage experiment of 6th WLS samples.

**Figure 13 materials-15-03319-f013:**
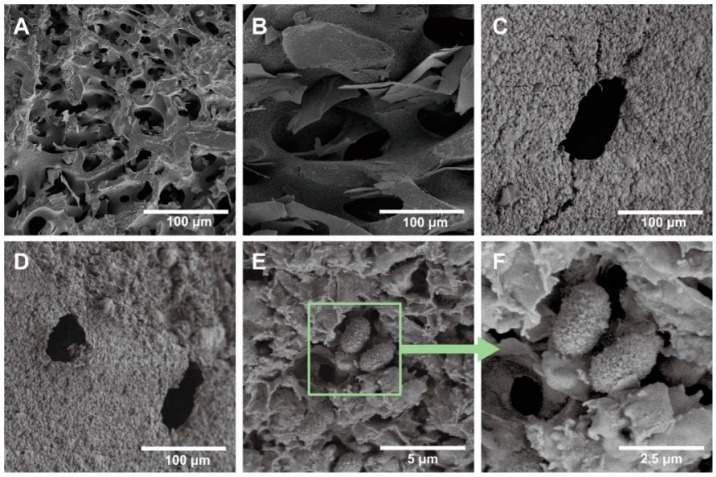
The SEM images of different samples. The cross-sectional images of (**A**) printed WES and (**B**) unprinted WES samples. Surface images of (**C**) printed WES and (**D**) unprinted WES samples. (**E**) The nitrifying bacteria in WLS, and the enlarged picture is shown in (**F**).

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
