# Peer review of "Waterborne Polyurethane Acrylates Preparation towards 3D Printing for Sewage Treatment"

_materials, 2022, doi:10.3390/ma15093319_

Round 1

Reviewer 1 Report

Please see attached pdf

Author Response

Thank you for the reviewer's time for improving the quality of our manuscript.

Here are our answers per your suggestions.

Point 1: In line 18, there is a typographical and grammatical mistake. Correct “are easily collapse” with “tend to easily collapse”.

Point 2: In lines 26-27 there is another grammatical mistake. After the word “practical”, it seems that a word is missing. Perhaps you can use the word “use” or “high potential”.

Point 3: In line 38 the word “fix” makes no sense in terms of correct use of English language. Instead, you can use the word “place”.

Point 4: In lines 44-46, even though the general meaning is correct, you should include a sentence indicating that “although the FDM and SLA methods are being used successfully in a number of applications [references] they are not suitable for printing bioactive inks because their printing process is not friendly to bacteria [6, 7].

Answer: We have revised the expressions in the revised manuscript per reviewer's suggestion. Thank you.

Point 4: The references that should be put in the yellow marked spot are at least the following:

  • doi: 10.5923/j.mechanics.20211001.02
  • doi: 10.3390/technologies9040091
  • doi: 10.3390/ma15072410

Answer: Thank you for your suggestion. Corresponding referenced had been cited in the revised manuscript.

Point 5: In lines 68-69 you are writing that “Compared to other curing methods, UV curing and ion-exchange curing together deliver a more bio-friendly fabrication process, facilitating the bacterium viability.” This sentence raises some serious questioning since UV Radiation is being used in sanitation devices with the exactly opposite expected result, that of killing bacteria and germs. Please rephrase or provide a more detailed explanation.

Answer: Thank you for your comment. In general, the sanitation devices employ the UV light with a wavelength ranged from 200-275 nm, which belong to UVC. Considering this, in this work, we adopted a UV curing device emits a UV light with a wavelength of 385 nm which belongs to UVA. It has been proved that the bacterial is survived under the exposure of UV light with this wavelength. [González, Lina M., Nikita Mukhitov, and Christopher A. Voigt. "Resilient living materials built by printing bacterial spores." Nature chemical biology 16.2 (2020): 126-133.] This wavelength is sufficient to initiate the free radical polymerization without killing bacteria and germs. Corresponding revisions have been made in the revised manuscript.

Point 6: In line 148 please write another process parameter, which is layer height. Which was the layer height of the 3d printed scaffolds? Was it the same at every different scaffold?

Answer: Thank you for your question. The layer thickness in this work was set as 0.58 mm. According to our previous experiments, the layer height is expected to be smaller than diameter of the syringe needle to achieve a high printing accuracy.

Point 7: In line 148 correct the abbreviation “CAE”. CAE software is used for simulation while CAD software is used for design. You should use the abbreviation “CAD”.

Answer: Thank you for pointing out our oversight. We revised the abbreviation to “CAD”.

Point 8: In line 150 you are mentioning the fact that you designed and printed some simple shaped scaffolds. Please show the different designs in a new figure. Also, mention other similar works existing in the literature like the following:
• doi: 10.3390/ma15072394
• doi: 10.3390/applmech2020018
• doi: 10.3390/ma15062305
• doi: 10.3390/ma15062170
• doi: 10.1007/s00170-015-7386-6
• doi: 10.1016/j.jbiomech.2004.12.022
• doi:10.1002/bit.22555
• doi: 10.3390/ma15041433

Answer: In this work we drew a cubic and a cylinder. The movement of the syringe is based on the G-code which is like FDM working principle. By varying different “infill ratio”, different lattice structures can be printed out. Corresponding illustration is as shown in below. (Detailed figure please check attached .pdf file. Thank you)

Fig. 3 A series of 3D scaffolding slice models with different infill rate and infill patterns were prepared by Simplify3D software.

This illustration is added in the revised manuscript.

Reviewer 2 Report

Dear Authors

The manuscript is novel; well presented and organized.

Some minor suggestions are given in the attached version of your manuscript that could improve your work.

In general, the article is well organised and structured. The authors used a variety of selected references that are novel and updated. The Figures are representative and mostly self-explained. Supplementary material is also a good addition to this article.  Great work. Furthermore, the ink developed in this work is ecofriendly and biocompatible that exhibits action towards bacteria. This is a good method for ink-printing and applications in biological systems.   As a summary, I recommend this paper to be published in your journal after addressing the comments/suggestions given in the attached file. It will be a good article for your readership. 

Please, take them into consideration.

Author Response

Thank you for pointing out our oversight in the manuscript. We’ve revised the grammatical errors in the revised manuscript in track change mode.

Reviewer 3 Report

The authors presented work entitled "Waterborne Polyurethane Acrylates Preparation toward 3D Printing for Sewage Treatment" is novel and benefits researchers in the area of 3D printing. The following points need to be addressed before final publication.

1) Line 43 - "3D printing technologies can process complex three-dimensional structures without geometry limitations". Provide reference

2) Line 76 - "This work aims to prepare an immobilized material with high mechanical properties". What mechanical properties? 

3) Line 84 - Why optimize process parameters?

4)  Line 109 - Why are authors PCL and PEG 109 (weight ratio = 19:1) chosen?

5) Line 114 - "decrease the viscosity of the system". Authors measured viscosity, if yes then write value.

6) Line 143 - Where is Figure S1? Provide the appropriate position.

7) Line 148 - CAE software. What CAE software?

8) Line 149 - Why do authors choose 20%, 30%, and 40% filling rates?

9) Line 180 - "To investigate the rheological properties of the ink during DIW printing, a simulation experiment including a high shear rate (80 1/s) and a low shear rate (0.01 1/s) were carried out". Show the simulation results.

10) Line 196 - "The mechanical properties of samples, including the tensile property of films and compression performance of printed scaffolds". What mechanical properties result?

11) Line 228-234 - What is the statement?

12) The figures are not appropriate position. 

13) Line 267-268  - "The curves shown in the figures demonstrate that all samples exhibit a shear-thinning behavior while the viscosity increases dramatically with the introduction of SA solution". Not clear

14) Line 274 - " From Fig. 3A, it should also be noted that". Wrong sentence.

Author Response

Responds to Reviewer 3

1) Line 43 - "3D printing technologies can process complex three-dimensional structures without geometry limitations". Provide reference

Answer: We’ve cited corresponding reference in the revised manuscript. Thank you.

Here are cited references:

  1. Jo, B. W. and Song, C. S. Thermoplastics and Photopolymer Desktop 3D Printing System Selection Criteria Based on Technical Specifications and Performances for Instructional Applications. Technologies 2021, 9 (4), 91, DOI, https://www.mdpi.com/2227-7080/9/4/91.
  2. al-Qarni, F. D. and Gad, M. M. Printing Accuracy and Flexural Properties of Different 3D-Printed Denture Base Resins. Materials 2022, 15 (7), 2410, DOI, https://www.mdpi.com/1996-1944/15/7/2410.

2) Line 76 - "This work aims to prepare an immobilized material with high mechanical properties". What mechanical properties?

Answer: Thank you. We’ve revised corresponding sentence to “This work aims to prepare an immobilized material with high tensile strength and strain, outstanding stability in practical usage, excellent biocompatibility to nitrifying bacteria, and a proper shear thinning behavior for DIW 3D printing.”

3) Line 84 - Why optimize process parameters?

Answer: Unlike traditional DLP and SLA 3D printing techniques, the parameters including extrusion pressure, the selection of syringe nozzle and proper layer thickness are significant parameters that affecting the printing accuracy. In our previous experiments, improper process parameters lead to a printing failure including overflow and incomplete construction. In the revised manuscript, we added “From the results of rheology study, proper extrusion pressure, diameter of the nozzle and the layer thickness were adjusted to print as-prepared WPUA to a bioreactor with a complex structure.”

4) Line 109 - Why are authors PCL and PEG (weight ratio = 19:1) chosen?

Answer: Due to the formation of hydrogen bonds between PCL and PEG, this formulation of PCL and PEG (19:1) provides the most suitable viscosity for DIW 3D printing along with a favorable rheology (shear thinning behavior).

5) Line 114 - "decrease the viscosity of the system". Authors measured viscosity, if yes then write value.

Answer: Thank you for your question. We’ve monitored the variation of the viscosity. Corresponding values had been added in the manuscript. We added “Subsequently, after increasing the temperature to 90°C, DMPA and MEK were added to decrease the viscosity of the system from 350.37 Pa s to 109.23 Pa s.”

6) Line 143 - Where is Figure S1? Provide the appropriate position.

Answer: The “Figures S1” is inserted in supplementary information. Here we paste the figure for your convenience. We added the relevant information in our revised manuscript “Supplementary Materials: The following supporting information can be downloaded at: www.mdpi.com/xxx/s1, Fig. S1: DIW machine used in this study; Fig. S2: (A) FTIR spectra of each reaction step of the synthetic WPUA. (B) The peak area changes of -NCO group during WPUA synthesis. (C) FTIR spectra of synthesized WPUAs. (D) The 1H NMR of WPUAs; Fig. S3: (A) TGA and (B) DSC curves of WPUA-E, WPUA-L, WES and WLS; Fig. S4: Intuitive diagram of 72 h dilution to 10-6 plate counting method. Table S1: Molar radio in WPUA; Table S2: Formulation of artificial sewage. (pH = 7.5); Table S3: The details for the determination of NH4+.”

(Attached figure can be seen in the PDF file)

Fig. S1 DIW machine used in this study

7) Line 148 - CAE software. What CAE software?

Answer: Thank you. The CAE (it should be called CAD software) used in this work is SolidWorks 2020 free trial version.

8) Line 149 - Why do authors choose 20%, 30%, and 40% filling rates?

Answer: Thank you for your question. In this work, we have tried filling rate other than 20%, 30% and 40%. According to the results, we found that when the filling rate is lower than 20%, the gaps between two drawing lines is too big to increase the ratio surface area efficiently. When the filling rate is higher than 40%, the drawing lines were too dense, which decrease the efficiency of mass transferring. Corresponding physical figures are shown in Figure 9A.

(Attached figure can be seen in the PDF file)

Fig. 9 (A) Photos of compression test samples. (B) The printing bioreactors with other infill patterns.

9) Line 180 - "To investigate the rheological properties of the ink during DIW printing, a simulation experiment including a high shear rate (80 1/s) and a low shear rate (0.01 1/s) were carried out". Show the simulation results.

Answer: Thank you for your question. During the process of DIW 3D printing, the extrusion of the bio-ink is realized by the gas pressure applied in the syringe. This work mechanism requires as-prepared bio-ink exhibit an obvious shear thinning behavior. The decrease of the viscosity when the pressure is applied makes the extrusion of the bio-ink fluently. While the high viscosity of the resin when the pressure was withdrawn improve the stability of the bio-ink. The simulation results are compiled in Figure 6. The blue dots showed in the lower half of the figure represent the “Shear Rate” numbers. Every 60 seconds, we changed the shear rate to evaluate the viscosity of the resin at this shear rate and time. We found that as-prepared WES and WLS responded fast to the variation of the shear rate, and no time latency can be observed. We improved the legends and description in the figure to make it easier to understand.

Fig. 6 Viscosity variation of WES and WLS in DIW 3D printing condition simulation.

10) Line 196 - "The mechanical properties of samples, including the tensile property of films and compression performance of printed scaffolds". What mechanical properties result?

Answer: The mechanical properties include tensile strength (results compiled in Figure 8(A)) compression performance (results compiled in Figure 8(B)). All corresponding results are compiled in section 3.3 Mechanical properties.

(Attached figure can be seen in the PDF file)

Fig. 8 (A) Tensile properties of WPUA-A, WPUA-L, WES, and WLS films. (B) Compression properties of WLS samples with 20%, 30%, and 40% filling ratio, respectively.

11) Line 228-234 - What is the statement?

Answer: Thank you for pointing out our oversight. We compiled corresponding text by the template downloaded from the website and forgot to delete the corresponding statement in the template. We’ve deleted corresponding texts in the revised manuscript.

12) The figures are not appropriate position.

Answer: In the revised manuscript, we have done the typesetting of the figures. Thank you.

13) Line 267-268 - "The curves shown in the figures demonstrate that all samples exhibit a shear-thinning behavior while the viscosity increases dramatically with the introduction of SA solution". Not clear

Answer: Thank you for your question. We replaced the original by “From Figure 3, It can be found that the introduction of the SA solution leads to an increase of the viscosity of the system. Also, a more obvious shear thinning behavior was observed.”

14) Line 274 - " From Fig. 3A, it should also be noted that". Wrong sentence.

Answer: Thank you for pointing out our oversight. We have modified corresponding expression in the revised manuscript.

Round 2

Reviewer 1 Report

I want to thank the authors for the corrections and the extensive editing they made. I strongly believe that the article is ready for publication now.

Reviewer 3 Report

Thank you for submitting the revised manuscript. Now the paper is suitable for publication.